# MSC−sEV Treatment Polarizes Pro−Fibrotic M2 Macrophages without Exacerbating Liver Fibrosis in NASH

**DOI:** 10.3390/ijms24098092

**Published:** 2023-04-30

**Authors:** Bin Zhang, Biyan Zhang, Ruenn Chai Lai, Wei Kian Sim, Kong Peng Lam, Sai Kiang Lim

**Affiliations:** 1Institute of Molecular and Cell Biology (IMCB), Agency for Science, Technology and Research (A*STAR), 61 Biopolis Drive, Proteos, Singapore 138673, Singapore; 2Singapore Immunology Network (SIgN), Agency for Science, Technology and Research (A*STAR), 8A Biomedical Grove, Immunos, Singapore 138648, Singapore; 3Department of Microbiology and Immunology, Yong Loo Lin School of Medicine, National University of Singapore, 5 Science Drive 2, Singapore 117545, Singapore; 4School of Biological Sciences, College of Science, Nanyang Technological University, 60 Nanyang Drive, Singapore 637551, Singapore; 5Department of Surgery, YLL School of Medicine, NUS, 5 Lower Kent Ridge Road, Singapore 119074, Singapore

**Keywords:** mesenchymal stem/stromal cells (MSCs), small extracellular vesicles (sEVs), fibrosis, non−alcoholic steatohepatitis (NASH), immunomodulation, M2 macrophage

## Abstract

Mesenchymal stem/stromal cell small extracellular vesicles (MSC−sEVs) have shown promise in treating a wide range of animal models of various human diseases, which has led to their consideration for clinical translation. However, the possibility of contraindication for MSC−sEV use is an important consideration. One concern is that MSC−sEVs have been shown to induce M2 macrophage polarization, which is known to be pro−fibrotic, potentially indicating contraindication in fibrotic diseases such as liver fibrosis. Despite this concern, previous studies have shown that MSC−sEVs alleviate high−fat diet (HFD)−induced non−alcoholic steatohepatitis (NASH). To assess whether the pro−fibrotic M2 macrophage polarization induced by MSC−sEVs could worsen liver fibrosis, we first verified that our MSC−sEV preparations could promote M2 polarization in vitro prior to their administration in a mouse model of NASH. Our results showed that treatment with MSC−sEVs reduced or had comparable NAFLD Activity Scores and liver fibrosis compared to vehicle− and Telmisartan−treated animals, respectively. Although CD163^+^ M2 macrophages were increased in the liver, and serum IL−6 levels were reduced in MSC−sEV treated animals, our data suggests that MSC−sEV treatment was efficacious in reducing liver fibrosis in a mouse model of NASH despite an increase in pro−fibrotic M2 macrophage polarization.

## 1. Introduction

Mesenchymal stromal/stem cells (MSCs) are a widely studied cell type that can be derived from various tissues in the body. The International Society for Cellular Therapy (ISCT) has defined these cells as being plastic−adherent when maintained in standard culture conditions, expressing CD105, CD73, and CD90, and lacking expression of CD45, CD34, CD14 or CD11b, CD79a or CD19, and HLA−DR surface molecules [1]. Additionally, MSCs should have the ability to differentiate into osteoblasts, adipocytes, and chondroblasts in vitro. MSCs have undergone numerous clinical trials for many diseases and have been found to be generally safe, with some functional or therapeutic improvements in many indications. While their initial use was predicated on their differentiation potential to replace damaged or diseased tissues, pre−clinical animal and clinical data suggest that MSCs primarily exert their therapeutic effects through secretion rather than differentiation [2,3]. We subsequently demonstrated that administration of MSC conditioned medium reduced infarct size in the myocardial infarction model and the active agent was larger than 1000 kD [4]. This agent was subsequently identified as the 80–1000 nm microvesicles by Cammussi and his group [5] and 100–130 nm MSC−sEVs by our group [6]. In 2017, Cammussi and his group reported that of the 80–1000 nm microvesicles, the smaller ~160 nm but not the larger ~215 nm microvesicles were therapeutic [7]. Therefore, it is now widely accepted that much of the therapeutic activity of MSCs could be attributed to sEVs in the 50–200 nm size range i.e., small EVs or sEVs [8], and the therapeutic activity of these vesicles was comparable to that of their parental MSCs [5,9,10,11].

While MSC−sEVs have much potential as therapeutic products, there are several challenges that need to be addressed. It is widely acknowledged that the therapeutic efficacy of MSC−sEV preparation is significantly influenced by the tissue source of the MSCs, such as bone marrow, adipose tissue, umbilical cord, or human embryonic stem cells, as well as the culture conditions of the MSCs. A recent meta−analysis of MSC−sEV proteomes deposited in public databases highlighted that while there is a common proteomic signature across MSC−sEV preparations, there are also significant differences between preparations that utilize different MSC sources and preparation protocols [12]. Such differences are reflected in the lack of consensus among different groups on the efficacious dosing range of exosomes for a disease target and the need in some labs to condition MSCs e.g., hypoxic vs normoxic for the production of therapeutic EVs. To circumvent this, MSC−sEV researchers representing four academic societies ISCT, ISEV, ISBT, and SOCRATES proposed that besides being defined by the process, MSC−sEV preparations should also be characterized using a matrix of quantifiable physical identity metrics that are common to all small (50–200 nm) MSC−derived lipid membrane vesicles [13]. This will differentiate MSC−sEV preparations with similar manufacturing processes and qualitative features, and help normalize different preparations.

Another key challenge in the path to clinical translation of MSC−sEVs is identifying contraindications to anticipate and reduce adverse events. A key feature of MSC−sEVs is their immune phenotype [14]. Like their producer cells, human MSC−sEVs are “immune privileged” and are routinely administered to fully immune competent animals without immune suppression to elicit therapeutic effects. As discussed earlier, MSC−sEVs which were derived from an immortalized MSC producer line, exhibit some proteomic differences with others derived from primary MSC cultures [12]. Their proteome includes >200 immunomodulatory proteins and exhibits many immune activities that were also displayed by MSCs [15]. For example, they do not express MHC class−I and II, or co−stimulatory molecules such as CD40, CD80, and CD86, attenuate inflammation by enhancing the secretion of anti−inflammatory cytokines, promoting Treg polarization inhibiting complement activation and polarizing THP−1 cells and primary mouse or human monocytes towards an M2−like macrophage phenotype with elevated expression of anti−inflammatory *IL10* and an attenuated expression of pro−inflammatory genes (e.g., *IL1β, IL6, TNFα, IL12p40*) [16,17,18,19], and inhibiting complement−activated neutrophil secretion of NETs and IL−17 [20,21]. MSC−sEVs were also reported to enhance healing of radiation−induced injury by mobilizing monocytes from spleen and bone marrow to promote neovascularization at the wound site [22].

While our MSC−sEV preparations have shown promising results in improving outcomes in osteochondral repair [23,24] and survival of allogeneic skin graft in mice [17], they could potentially aggravate other diseases and may exacerbate disease progression or contribute to tissue damage such as promoting tumor growth and metastasis [25]. However, the role of macrophages in many pathologic settings is highly complex and dependent on a highly dynamic balance between different macrophage subsets. For example, the M1 and M2 subsets of macrophages played highly dynamic roles in the pathologic progression of non−alcoholic steatohepatitis (NASH) [26]. It has been proposed that in established organ fibrosis such as liver fibrosis, M1 macrophages are anti−fibrotic by actively phagocytosing debris and degrading connective tissue while M2 macrophages are pro−fibrotic by secreting tissue remodeling factors such as fibronectin−1, matrix associated protein betaIG−H3, coagulation factor XIII, tissue−type plasmin activator and insulin−like growth factors (IGF) [27]. Furthermore, the blockade of IL−6R signaling has been implicated in increased liver fibrosis [28,29] and enhanced hepatic steatosis while reducing liver injury [30] suggesting IL−6 signaling may attenuate liver fibrosis. Therefore, MSC−sEVs could potentially aggravate fibrosis in established organ fibrosis through enhanced polarization of M2 macrophages and reduced IL−6 production.

To test this possibility, we demonstrate here that our MSC−sEV preparation can polarize human M0 macrophage to M2. We then administered the preparations to the STAM™ mouse model [31] to test if the preparation could exacerbate NASH and liver fibrosis with a concomitant increase in CD163^+^ M2 macrophage and a decrease in IL−6. Although several independent research groups have reported that their MSC−sEV preparations were efficacious in reducing hepatic injury and fibrosis in different mouse models of NASH [32,33,34], the potential of their MSC−sEV preparations to polarize M2 macrophages and reduce IL−6, or the effects of this potential on the development of fibrosis has not been reported.

## 2. Results

### 2.1. MSC−sEV Preparations Can Polarize M0 to M2 but Not M1 Macrophages

To confirm if our MSC−sEV preparation can polarize human M0 macrophages to either M1 or M2, peripheral blood mononuclear cells (PBMCs) were cultured in the presence of human M−CSF for six days to activate the monocytes to M0 macrophages. The M0 macrophages were then polarized to M1 or M2 macrophages by culturing in IFNγ + LPS or IL−4, respectively, for 24 h or in the presence of 0.1, 1, 10, or 30 µg/mL MSC−sEVs as described in Materials and Methods. The cells were then harvested and analyzed for M1 or M2 surface markers. Here we found that MSC−sEVs significantly polarized M0 to M2 (CD68^+^ CD206^hi^ CD163^hi^ macrophages, Figure 1) but not M1(CD68^+^ PDL1^hi^ CD38^hi^ macrophages, Figure 2). Interestingly, the level of M2 polarization was statistically similar at all concentrations of MSC−sEVs. (Figure 1B).

### 2.2. MSC−sEV Preparations Suppress Fibrosis in a Mouse Model of NASH

To assess whether MSC−sEV preparations have the potential to exacerbate NASH by polarizing M2 macrophages, we administered 1 and 10 μg per STAM™ mouse via intraperitoneal (IP) injection, as described in Materials and Methods (Figure 3). To serve as a positive control, we also administered Telmisartan, an angiotensin II receptor blocker (ARB) that has been reported to reduce fibrosis in NASH [35] (Figure 3).

Compared to the naïve (normal) mice, the induction of NASH caused a reduction in body weight and an increase in liver weight (Figure 4A). Telmisartan treatment caused a statistically significant reduction in total body weight and liver weight in this mouse model, consistent with the previous report [36]. However, there were no significant differences in mean body or liver weight between the vehicle group and the two MSC−sEV groups during the treatment period (Figure 4A), and consequently, there were also no significant differences in mean liver−to−body weight ratio among these groups (Figure 4A).

We assessed liver damage by measuring plasma ALT levels, a common diagnostic marker. All NASH animals showed elevated ALT levels, with no statistically significant differences among the groups, including the vehicle control group (Figure 4B). Liver triglyceride levels, which have a strong positive correlation with NAFLD [37], were highly elevated in all NASH animals compared to the normal animals, but there were no statistical differences among the treatment groups, including the vehicle control group (Figure 4C).

Taken together, these data indicate that MSC−sEVs did not exacerbate liver damage or liver triglyceride levels in the NASH animals.

The livers of both normal and NASH mice were examined by staining with H & E and scored for the presence of micro− and macrovesicular fat deposition (steatosis), hepatocellular ballooning, and inflammatory cell infiltration, as defined in Table 1 (Figure 5A). The cumulative score for these three parameters constituted the NAFLD activity score (NAS), which was higher in all NASH mice than in normal mice. The vehicle group had the highest score, while both the MSC−sEV and Telmisartan treatment groups had statistically significant lower scores than the vehicle group (Figure 5B). There was no difference in NAS between the 1 µg and 10 µg MSC−sEV treatment groups, but both had a higher score than the Telmisartan group (Figure 5B). Both doses of MSC−sEVs significantly reduced NAS compared to the vehicle group (Figure 5B).

There was a higher NAS observed in the NASH mice compared to the normal mice, which was consistent with pathological collagen deposition as indicated by Sirius Red staining, and a concurrent increase in fibrosis area (Figure 5C). However, in all treatment groups, there was an attenuation of this increase, although it was not statistically significant for the 1 µg MSC−sEV treatment group. This lack of statistical significance is likely a statistical anomaly because the fibrotic area in the 1 µg MSC−sEV treatment group did not significantly differ from that in the 10 µg MSC−sEV and Telmisartan treatment groups (Figure 5D).

### 2.3. MSC−sEV Preparations Increase Anti−Inflammatory M2 Macrophages

After demonstrating in vitro that MSC−sEVs can polarize profibrotic M2 macrophages, we hypothesized that the administration of MSC−sEVs to a mouse model of liver fibrosis would increase liver fibrosis. However, contrary to our expectations, we did not observe an increase in liver fibrosis. To determine if the in vitro potential of MSC−sEVs was manifested in this model, we assessed the relative abundance of CD163^+^ M2 macrophages in liver sections.

We observed that the MSC−sEV−treated groups displayed significantly higher infiltration of CD163^+^ cells in the livers of mice treated intraperitoneally with 1 µg and 10 µg MSC−sEVs (1.5 ± 0.3 vs. 1.0 ± 0.3, *p* = 0.0147; 1.8 ± 0.3 vs. 1.0 ± 0.3, *p* = 0.0005), relative to the vehicle control (Figure 6A,B). Additionally, we analyzed the plasma for IL−6 using a commercially available cytokine array panel consisting of 32 cytokines, including Eotaxin, G−CSF, GM−CSF, IFNγ, IL−1α, IL−1β, IL−2, IL−3, IL−4, IL−5, IL−6, IL−7, IL−9, IL−10, IL−12p40, IL−12p70, IL−13, IL−15, IL−17A, IP−10, KC, LIF, LIX, MCP−1, M−CSF, MIG, MIP−1α, MIP−1β, MIP−2, RANTES, TNFα, and VEGF−A (Figure 7).

We found that IL−6 levels were reduced in the plasma of mice treated with one and 10 µg MSC−sEVs relative to those treated with vehicle and Telmisartan, and the reductions for IL−6 were statistically significant at *p* ≤ 0.05 but not for mice treated with 1 µg MSC−sEVs (Figure 7). The latter could be a statistical anomaly as no significant differences were observed in IL−6 levels between the mice treated with 10 µg MSC−sEVs and 1 µg MSC−sEVs. These results suggest that MSC−sEVs can polarize M2 macrophages and reduce IL−6 levels in a mouse model of liver fibrosis.

## 3. Discussion

In this study, we investigated the effects of MSC−sEVs on NASH mice to assess their potential in mediating the polarization of pro−fibrotic M2 macrophages and reducing IL−6 levels, and to determine if they exacerbate liver fibrosis. Despite an increase in M2 macrophages in the liver and a decrease in plasma IL−6 in MSC−sEV−treated mice, we found that MSC−sEVs did not worsen liver function or triglyceride levels. In fact, we observed that MSC−sEVs reduced NAS, liver fibrosis, and collagen deposits, which is consistent with previous reports [32,33,34]. These findings suggest that our MSC−sEV preparation was not pro−fibrotic, despite increasing the number of pro−fibrotic M2 macrophages in the liver and reducing plasma IL−6 levels.

It is important to note that our results do not definitively rule out the possibility that pro−fibrotic M2 macrophages could exacerbate NASH. Rather, our observations should be interpreted within the broader context of the complex immune pathology of NASH and the many immune−modulating activities of MSC−sEVs.

Although tissue macrophages are well studied and considered central to the inflammatory pathology of NASH [38], there is growing evidence implicating T and B cells in NASH. Various subsets of T cells and B cells, including CD8^+^ T cells, CD4^+^ T helper cells, γδ T cells, NKT cells, MAIT cells, Tregs, Bregs, antibody−secreting plasma cells, and memory B cells have been shown to contribute to NASH pathogenesis. However, the complex interactions and intersecting roles of these immune cell subsets remain poorly understood [39,40]. In addition to adaptive immune cells, innate immune cells such as neutrophils, and soluble humoral factors like complements, also play a role in the immune pathology of NASH [41,42].

Therefore, to gain a better understanding of how MSC−sEVs function in NASH immune pathogenesis, it is crucial to understand the interplay between different immune compartments and the various immune−modulating activities of MSC−sEVs. Previous studies have demonstrated that our MSC−sEV preparations could activate TLR4 and stimulate an anti−inflammatory response in human and mouse primary monocytes [17]. Furthermore, in vitro experiments have shown that these MSC−sEVs can induce naïve CD4^+^ T cells into CD4^+^CD25^+^Foxp3^+^ Tregs in the presence of allogenic CD11c^+^ APCs [16]. In vivo studies have also demonstrated that MSC−sEVs can increase Tregs in mice with an allogenic skin graft or GVHD [16,17]. This suggests that immune cells from immunologically challenged animals are primed to polarize into regulatory cell types by MSC exosomes [17]. Furthermore, MSC−sEVs can modulate soluble humoral factors and innate immune cells by inhibiting the formation of the terminal complement complex through CD59 present in MSC−sEVs [18]. This, in turn, inhibits complement−activated neutrophils, reducing NETS and IL−17 secretion in vitro [21] and in vivo [20].

This study highlights the limitation of extrapolating linearly from a single modulating activity of MSC−sEVs when evaluating the potential effects of MSC−sEVs on disease progression, especially for complex diseases such as NASH. We had hypothesized that the use of MSC−sEVs would be harmful in the treatment of NASH on the basis of a single modulating activity of MSC−sEVs i.e., the polarization of pro−fibrotic M2 macrophages in vitro. However, the in vivo experiments showed that administration of MSC−sEVs to NASH mice increased pro−fibrotic M2 macrophages but did not exacerbate liver fibrosis, as predicted. Instead, there was a reduction in fibrosis. Furthermore, the study notes that there is no apparent relationship between sEV concentration and the magnitude of biological effects in both in vitro and in vivo studies, suggesting that factors other than sEV concentration may be more influential. Overall, this study emphasizes the need for a comprehensive and nuanced understanding of the mechanisms underlying the therapeutic effects of MSC−sEVs in complex diseases, such as NASH, in order to develop effective treatment strategies. These strategies will likely involve a more holistic integration of multiple MSC−sEV attributes within the underlying pathology of the target disease.

## 4. Materials and Methods

### 4.1. Culture of MSCs and Preparation of MSC−sEVs

Immortalized E1−MYC 16.3 human ESC−derived mesenchymal stem cells were cultured in DMEM with 10% fetal calf serum as previously described [43]. For MSC−sEV preparation, the conditioned medium was prepared by growing 80% confluent cells in a chemically defined medium for three days as previously described [6,44,45]. The defined medium was prepared as follows: 480 mL DMEM (31053, Thermo Fisher, Waltham, MA, USA), 5 mL NEAA (11140−050, Thermo Fisher, Waltham, MA, USA), 5 mL L Glutamine (25030−081, Thermo Fisher, Waltham, MA, USA), 5 mL Sodium Pyruvate (11360, Thermo Fisher, Waltham, MA, USA), 5 mL ITS−X (51500−056, Thermo Fisher, Waltham, MA, USA), 0.5 mL 2−ME (21985−02, Thermo Fisher, Waltham, MA, USA). This was supplemented with 0.1 mL bFGF (0.5 ng/μL 0.2% BSA in PBS (+) and 0.005 mL PDGF (100 ng/μL PBS (+)). These latter components were obtained as follows: Bovine Serum Albumin or BSA (A9647, Sigma−Aldrich, St. Louis, MO, USA), PDGF (100−00 AB CYTOLAB), bFGF (13256−029, Thermo Fisher, Waltham, MA, USA) and PBS (+) (14040−133, Thermo Fisher, Waltham, MA, USA). The conditioned medium (CM) was size−fractionated by tangential flow filtration and then concentrated 50× using a membrane with a molecular weight cut−off (MWCO) of 100 kDa (Sartorius, Gottingen, Germany). The MSC−sEV preparation was assayed for protein concentration using a Coomassie Plus (Bradford) Assay Kit (ThermoFisher Scientific, Waltham, MA, USA). Only batches of sEV determined by Nanoparticle tracking analysis on a ZetaView instrument (Particle Matrix GmbH, Germany) to have 1.46 × 10^11^ ± 2.43 × 10^10^ particles per ug protein and particle modal size of 138.62 ± 4.45 nm using the parameters (sensitivity = 90, shutter = 70, frame rate = 30, min brightness = 25, min area = 5, max area = 1000) were used for this study. In addition, preparations must express CD81 and CD73 as determined by western or ELISA. The MSC−sEV preparations were filtered with a 0.22μm filter (Merck Millipore, Billerica, MA, USA) and stored in a −80 °C freezer.

### 4.2. Polarization of Macrophages by MSC−sEVs

Human blood samples from apheresis cones were obtained for this study with approval from the Centralised Institutional Research Board of the Singapore Health Services. Primary human monocytes were obtained from human peripheral blood mononuclear cells (PBMCs) isolated from human blood apheresis cone provided by the Health Sciences Authority under the project number 201306−04. Briefly, peripheral blood mononuclear cells (PBMCs) from an apheresis cone were isolated by density centrifugation with Ficoll−Paque density gradient media. Primary monocytes were isolated from PBMCs by negative selection using a monocyte isolation kit (Cat No: #19359, Stem Cell Technologies, Vancouver, BC, Canada). Monocytes were plated at a concentration of 1 × 10^6^ cells/mL in plain RPMI in 6−well culture plate for 2 h to allow cells to adhere. Subsequently, plated monocytes were cultured in complete media (RPMI + 10% FBS, 100 units/mL Penicillin, and 100 µg/mL Streptomycin) with human Macrophage colony−stimulating factor (M−CSF) for polarization to M0 macrophages for 6 days. Fresh media with M−CSF was added on the third day. On day 6, monocyte−derived macrophages were treated for 24 h with 0.1, 1, 10, or 30 µg/mL MSC−sEVs in the presence of 20 ng/mL IFNγ + 100 ng/mL LPS (M1) (IFNγ: R & D systems, Minneapolis, MN, USA; LPS: Sigma Aldrich, St. Louis, MO, USA) or 20 ng/mL IL−4 (M2) (R & D systems, Minneapolis, MN, USA). Surface markers characteristic of M1 and M2 macrophages were examined by flow cytometry analysis. Twenty−four hours after treatment with MSC−sEVs, M1 or M2 polarizing factors, cells were removed from the plate with 0.05% Trypsin−EDTA (Gibco, Grand Island, NY, USA) followed by gentle scraping. Cells were washed twice in 2% FBS in PBS before incubation on ice for 30 min with fixable viability dye (eBioscience, San Diego, CA, USA) and anti−human CD206−BV421, CD163−PerCP Cy5.5, CD38−PE, PDL1−PECy7 and CD68−APC fluorescent conjugated antibodies (BD Biosciences, San Jose, CA, USA). After staining, cells were washed with 2% FBS in PBS before analysis by flow cytometry (BD FACSymphony™ A5.2 Cell Analyzer). Flow cytometry data were analyzed using FlowJo™ Software (Version 10).

### 4.3. HFD−Induced NASH Model

This study was performed by SMC Laboratories, Inc. 2−16−1 Minami−Kamata Ota−City, Tokyo 144−0035, Japan under IACUC no: S190. Fourteen−day−pregnant female C57BL/6J mice were obtained from Japan SLC, Inc. The Animal Experimentation Committee approved the use of all animals in this study (Approval No. S190, 12 October 2020), and all animals were cared for in accordance with established guidelines. These guidelines included the Act on Welfare and Management of Animals (Ministry of the Environment, Act No. 105 of 1 October 1973), the Standards Relating to the Care and Management of Laboratory Animals and Relief of Pain (Notice No. 88 of the Ministry of the Environment, 28 April 2006), and the Guidelines for Proper Conduct of Animal Experiments (Science Council of Japan, 1 June 2006). Briefly, 2 days old male mice were given a single subcutaneous injection of 200 µg streptozotocin (STZ, Sigma−Aldrich, USA) solution and fed a high−fat diet (HFD, 57 kcal% fat, Cat# HFD32, CLEA Japan, Inc., Tokyo, Japan) at 4 weeks to induce NASH [46,47,48]. Mice were identified by ear punch and randomized into 4 groups of 8 mice at 6 weeks of age based on their body weight. Five normal mice served as the normal control group. The MSC−sEVs were prepared as above and the telmisartan (purchased from Boehringer Ingelheim GmbH, Germany) was freshly prepared prior to administration as a positive control. One tablet of telmisartan was transferred into a mortar and triturated using pestle by adding RO water gradually to get a 1 mg/mL of homogeneous suspension. MSC−sEVs were administered intraperitoneally at doses of 1 and 10 μg/50 μL/mouse every other day, PBS was administered as per MSC−sEVs as a vehicle control and telmisartan was administered orally at a dose of 10 mg/10 mL/kg daily [35]. The viability, clinical signs (lethargy, twitching, labored breathing), and behavior were monitored daily. The body weight was recorded daily before the treatment. Mice were observed for significant clinical signs of toxicity, moribundity, and mortality before administration. The animals were sacrificed at 9 weeks of age by exsanguination through direct cardiac puncture under isoflurane anesthesia (Pfizer Inc., Chesterfield, MO, USA). The non−fasting blood samples were collected in polypropylene tubes with anticoagulant (Novo−Heparin, Mochida Pharmaceutical Co. Ltd., Tokyo, Japan) and centrifuged at 1000× *g* for 15 min at 4 °C. The supernatants were collected and the ALT levels were measured by FUJI DRI−CHEM 7000 (Fujifilm Corporation, Tokyo, Japan). Other plasma was stored at −80 °C until further use. The liver samples were collected, photographed, and weighed. The liver−to−body weight ratio was calculated. Some liver specimens were snap−frozen in liquid nitrogen and stored at −80 °C for triglyceride biochemistry analysis. Some specimens were fixed in Bouin’s solution (Sigma−Aldrich Japan, Tokyo, Japan) for 24 h. After fixation, these specimens proceeded to paraffin embedding for HE and Sirius red−staining. Other liver specimens were stored at −80 °C embedded in Optimal Cutting Temperature (O.C.T., Sakura Finetek Japan, Tokyo, Japan) compound for further analysis.

### 4.4. Liver Triglyceride Measurement

Liver total lipid extracts were obtained by Folch’s method [49]. Liver samples were homogenized in chloroform−methanol (2:1, *v*/*v*) and incubated overnight at room temperature. After washing with chloroform−methanol−water (8:4:3, *v*/*v*/*v*), the extracts were evaporated to dryness, and dissolved in isopropanol. Liver triglyceride was measured by Triglyceride *E*−test (FUJIFILM Wako Pure Chemical Corporation, Osaka, Japan).

### 4.5. Liver Histological Analysis

Liver sections were prepared and stained with H & E to determine micro− and macro−vesicular fat deposition (steatosis), hepatocellular ballooning, and lobular inflammatory cell infiltration according to the criteria of Kleiner histological analyses to determine NAS and Sirius red or/and Masson’s Trichrome staining for collagen deposition and fibrosis area.

For HE staining, sections will be cut from paraffin blocks of liver tissue prefixed in Bouin’s solution using the rotary microtome (Leica Microsystems) and stained with Lillie−Mayer’s Hematoxylin (Muto Pure Chemicals Co., Ltd., Tokyo, Japan) and eosin solution (Wako Pure Chemical Industries). NAS will be calculated according to the criteria of Kleiner [50] as shown in Table 1. For scoring of NAS, bright field images of HE−stained sections will be captured using a digital camera (DFC295; Leica, Germany) at 50−and 200−fold magnifications. Steatosis score in 1 section/mouse (representative 1 field around the central vein at 50−fold magnification), inflammation score in 1 section/mouse (representative 1 field around the central vein at 200−fold magnification), and ballooning score in 1 section/mouse (representative 1 field around the central vein at 200−fold magnification) will be estimated.

To visualize collagen deposition, Bouin’s fixed liver sections were stained using picro−Sirius red solution (Waldeck, Germany). Briefly, sections were deparaffinized and hydrophilized with xylene, 100–70% alcohol series, and RO water, and then treated with 0.03% picro−Sirius red solution (Cat No.: 1A−280) for 60 min. After washing through 0.5% acetic acid solution and RO water, stained sections were dehydrated and cleared with 70–100% alcohol series and xylene, then sealed with Entellan^®^ new (Merck, Darmstadt, Germany) and used for observation. For quantitative analysis of fibrosis area, bright field images of Sirius red−stained sections were captured around the central vein using a digital camera (DFC295; Leica, Wetzlar, Germany) at 200−fold magnification, and the positive areas in 5 fields/section were measured using ImageJ software (National Institute of Health, Bethesda, MD, USA).

### 4.6. Immunohistochemistry Staining for CD163

As a marker of M2 macrophages, CD163 was assessed [51] by standard immunohistochemistry (IHC). For NASH liver IHC, the sections at 6 μm cut from O.C.T. blocks will be fixed in acetone. Endogenous peroxidase activity will be blocked using 0.3% H_2_O_2_ for 5 min. The slides will then be incubated with antigen retrieval reagent (RM102−H, LSI Medience Corporation, Tokyo, Japan) for 10 min at 121 °C. The sections will be incubated with anti−CD163 antibodies (clone no. EPR19518, Rabbit monoclonal, Abcam) at 4 °C overnight. After incubation with secondary antibody (VECTASTAIN^®^ Elite ABC−HRP Kit, Vector Laboratories, Inc., Newark, CA, USA), enzyme−substrate reactions will be performed using 3, 3′−diaminobenzidine/H_2_O_2_ solution (Nichirei Bioscience Inc., Tokyo, Japan). To observe the CD163−positive areas, bright field images of CD163−immunostained sections will be captured in 3–5 fields including positive areas per section using a digital camera (DFC295) at 200−fold magnifications. For quantitative analysis of CD163, positively−stained cells were counted in three randomly−selected fields at 200X magnification and expressed as the mean number of cells per high power field (cells/HPF).

### 4.7. Mouse Plasmatic Cytokine/Chemokine Array

This assay was performed by Eve Technologies 3415A—3 Ave., N. W. Calgary, AB Canada T2N 0M4. The mouse plasmatic levels of Eotaxin, G−CSF, GM−CSF, IFNγ, IL−1α, IL−1β, IL−2, IL−3, IL−4, IL−5, IL−6, IL−7, IL−9, IL−10, IL−12p40, IL−12p70, IL−13, IL−15, IL−17A, IP−10, KC, LIF, LIX, MCP−1, M−CSF, MIG, MIP−1α, MIP−1β, MIP−2, RANTES, TNFα, VEGF−A were measured by Luminex technology (Mouse Cytokine/Chemokine 32−Plex Discovery Assay^®^ Array). The samples were diluted according to the company’s instructions. The results were analyzed using Student’s *t*−test. *p* values < 0.05 were considered statistically significant.

### 4.8. Statistical Tests

Statistical analyses were performed using Bonferroni Multiple Comparison Test on GraphPad Prism 6 (GraphPad Software Inc., San Diego, CA, USA). *p* values < 0.05 were considered statistically significant. The average value and standard deviation of each group were calculated by the individual animal in the group. A trend or tendency was assumed when a one−tailed *t*−test returned *p* values < 0.1. Results were expressed as mean ± SD.

## Figures and Tables

**Figure 1 ijms-24-08092-f001:**
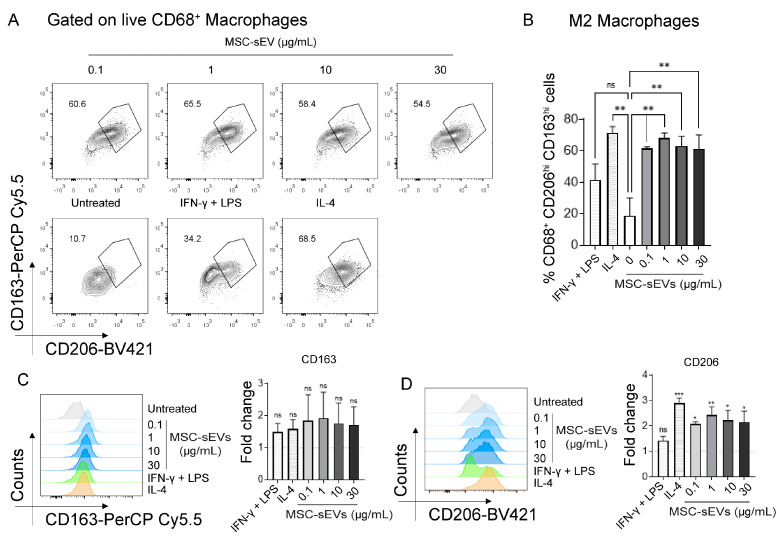
Representative flow cytometry plots (**A**), and summary bar graph (**B**) of CD206^hi^ CD163^hi^ macrophages in six−day cultured monocyte−derived macrophages (M−CSF treated) treated for 24 h with 0.1, 1, 10, or 30 µg/mL MSC−sEVs compared to negative control (untreated) and positive control M1 (IFNγ + LPS) and M2 polarized (IL−4) macrophages. Representative histograms and summary bar graph of expression of (**C**) CD163 or (**D**) CD206 on macrophages in six−day cultured monocyte−derived macrophages (M−CSF treated) treated for 24 h with 0.1, 1, 10, or 30 µg/mL MSC−sEVs compared to negative control (untreated) and positive control M1 (IFNγ + LPS) and M2 polarized (IL−4) macrophages. The fold change was determined by normalization to the ‘‘untreated’’ negative control. Each data point was expressed as a mean (±SD) of four independent assays performed in triplicate, *** *p* < 0.001, ** *p* < 0.01, * *p* < 0.05. The “ns” represents “not significant”.

**Figure 2 ijms-24-08092-f002:**
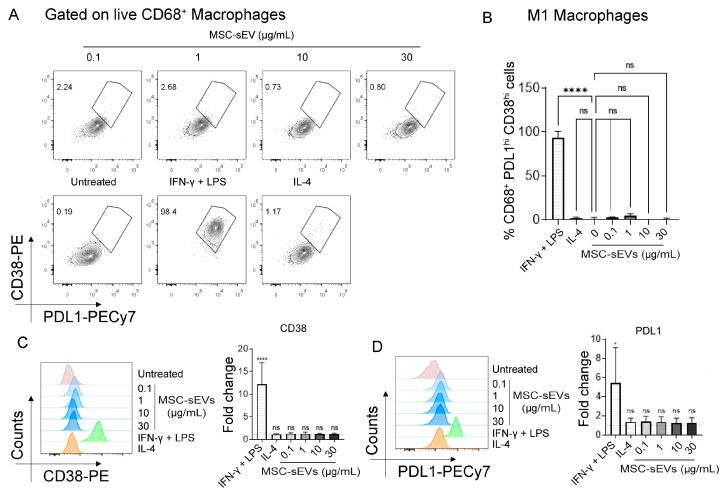
Representative flow cytometry plots (**A**) and summary bar graph (**B**) of PDL1^hi^ CD38^hi^ macrophages in six−day cultured monocyte−derived macrophages (M−CSF treated) treated for 24 h with 0.1, 1, 10, or 30 µg/mL MSC−sEVs compared to negative control (untreated) and positive control M1 (IFNγ + LPS) and M2 polarized (IL−4) macrophages. Representative histograms and summary bar graph of the expression of (**C**) CD38 or (**D**) PDL1 on macrophages in six−day cultured monocyte−derived macrophages (M−CSF treated) treated for 24 h with 0.1, 1, 10, or 30 µg/mL MSC−sEVs compared to negative control (untreated) and positive control M1 (IFNγ + LPS) and M2 polarized (IL−4) macrophages. The fold change was determined by normalization to the ‘‘untreated’’ negative control. Each bar was expressed as a mean (±SD) of four independent assays performed in triplicate, **** *p* < 0.0001, * *p* < 0.05. The “ns” represents “not significant”.

**Figure 3 ijms-24-08092-f003:**
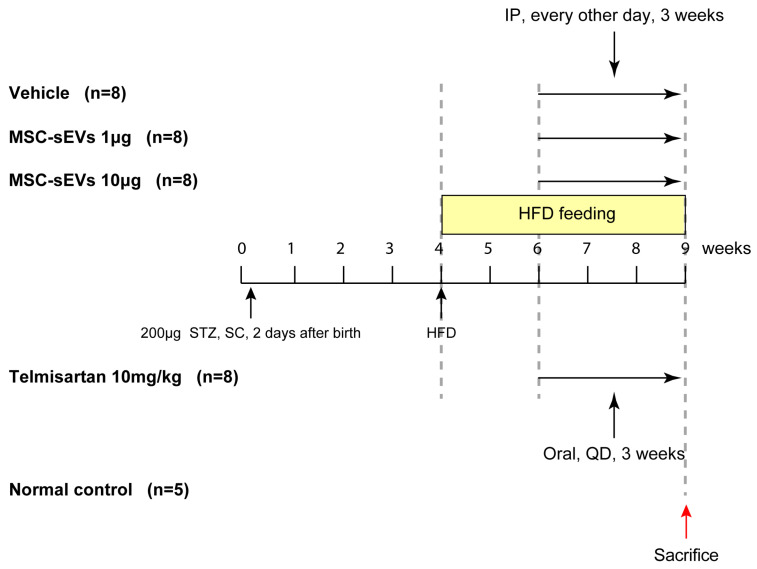
Summary of administrations in the HFD induced NASH mouse model. A single SC of 200 µg STZ was performed on mice at two days after birth and mice were fed with HFD after four weeks of age. Mice were randomized into four groups of eight mice at six weeks of age. Five normal mice were fed with normal diet as the normal control group. The MSC−sEVs were administered intraperitoneally at doses of one and 10 μg/50 μL/mouse every other day, PBS was administered as per MSC−sEVs as a vehicle control and telmisartan was administered orally at a dose of 10 mg/10 mL/kg daily as a positive control. The animals were sacrificed at nine weeks of age. The red arrow indicates the study termination day. STZ: streptozotocin; SC: subcutaneous injection; HFD: high fat diet; Oral: oral administration; IP: intraperitoneal administration; QD: once daily.

**Figure 4 ijms-24-08092-f004:**
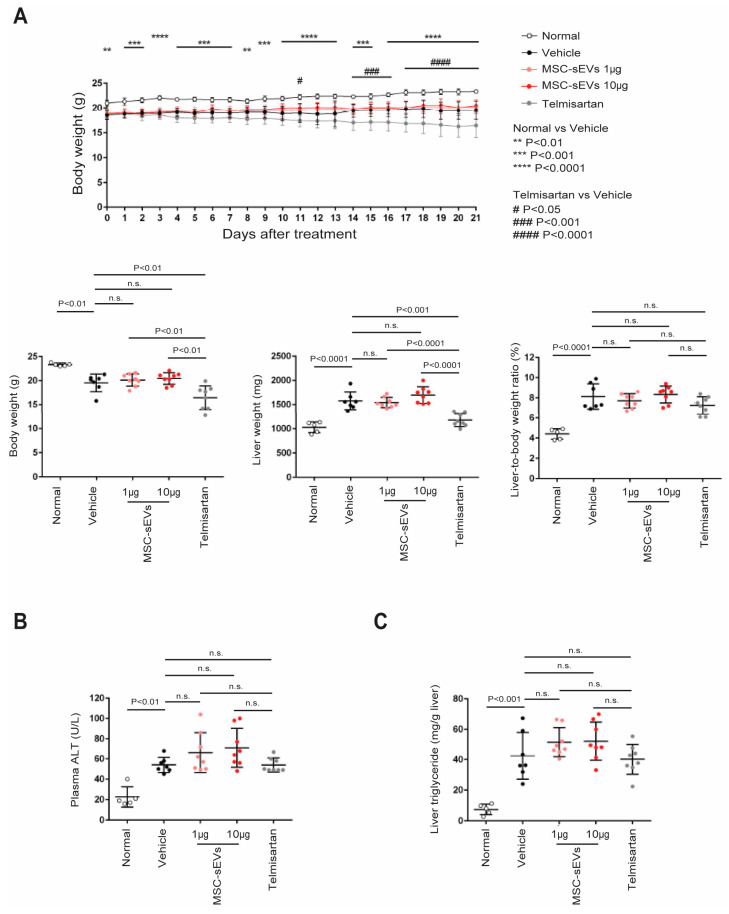
Evaluation of body/liver weight and measurement of plasma ALT/liver triglyceride in NASH mouse model. Body weight was recorded daily after the treatment and animals were sacrificed at nine weeks of age. The liver samples were collected, photographed, and weighed. The liver−to−body weight ratio was calculated (**A**). The plasma samples were collected, and ALT levels were measured by a blood chemistry analyzer (**B**). Some liver specimens were homogenized in chloroform−methanol (2:1, *v*/*v*) and incubated overnight at room temperature. After washing with chloroform−methanol−water (8:4:3, *v*/*v*/*v*), the extracts were evaporated to dryness, and dissolved in isopropanol for triglyceride measurement by Triglyceride *E*−test (**C**).

**Figure 5 ijms-24-08092-f005:**
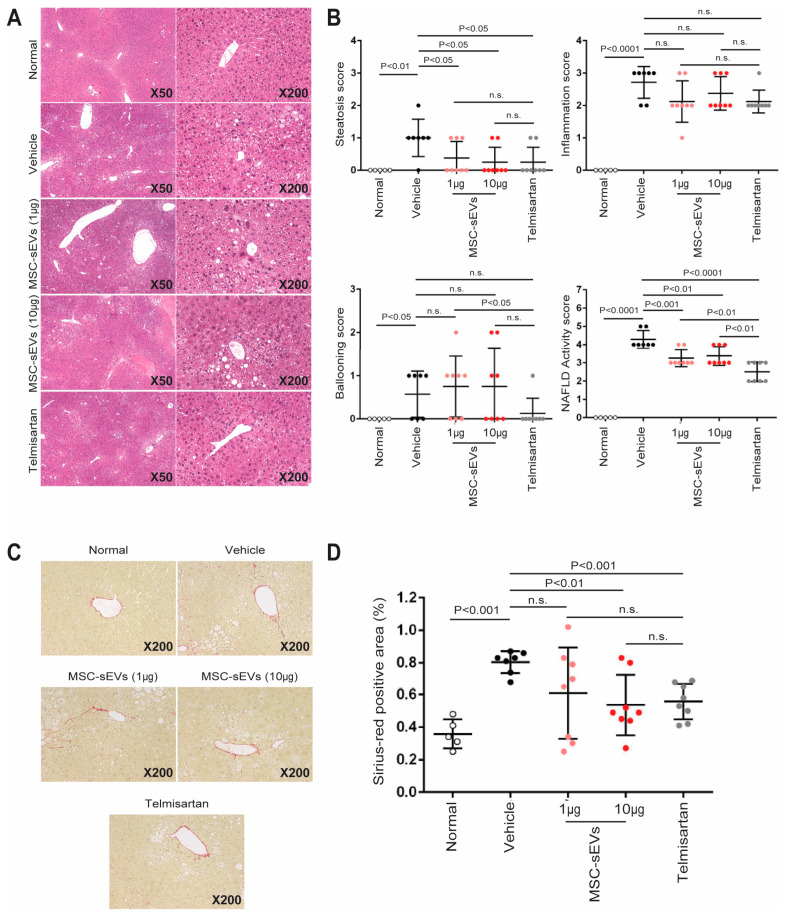
Evaluation of anti−fibrosis mediated by MSC−sEVs in NASH mouse model. (**A**) Representative photomicrographs of HE−stained liver sections are shown. The sections were cut from paraffin blocks of liver tissue prefixed in Bouin’s solution and stained with Lillie−Mayer’s Hematoxylin and eosin solution. (**B**) NAS was calculated according to the criteria of Kleiner. The bright field image of HE−stained sections was captured using a digital camera at 50−and 200−fold magnifications. Steatosis score in one section/mouse (representative one field around the central vein at 50−fold magnification), inflammation score in one section/mouse (representative one field around the central vein at 200−fold magnification), and ballooning score in one section/mouse (representative one field around the central vein at 200−fold magnification) were estimated. (**C**) Representative photomicrographs of Sirius red−stained liver sections are shown. Bouin’s fixed liver sections were stained using picro−Sirius red solution to visualize collagen deposition. (**D**) The quantitative analysis of fibrosis area. The bright field images of Sirius red−stained sections were captured around the central vein using a digital camera at 200−fold magnification, and the positive areas in five fields/section were measured using ImageJ software.

**Figure 6 ijms-24-08092-f006:**
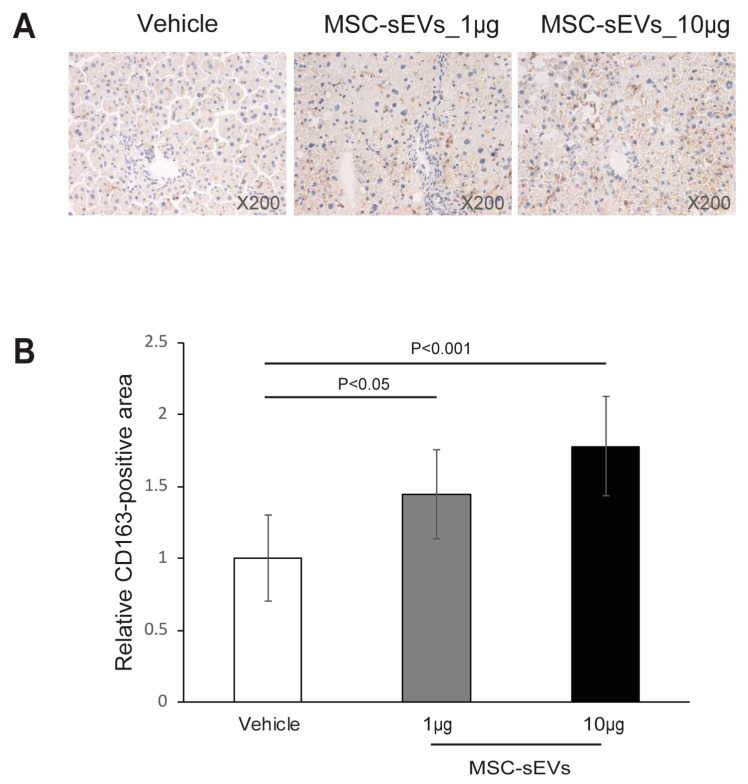
Effects of MSC−sEVs on M2 macrophage infiltration. Representative photomicrographs of CD163−immunohistochemistry−stained liver ((**A**), upper panel) sections of NASH mice are shown. For quantitative analysis of CD163−positive areas, bright field images of CD163−immunostained sections were captured using a digital camera at 200−fold magnification, and CD163^+^ cells were counted per high power field (HPF) in liver sections ((**B**), lower panel), and the positive areas in five fields/section were measured using ImageJ software. Data represent mean ± SD. compared to vehicle control.

**Figure 7 ijms-24-08092-f007:**
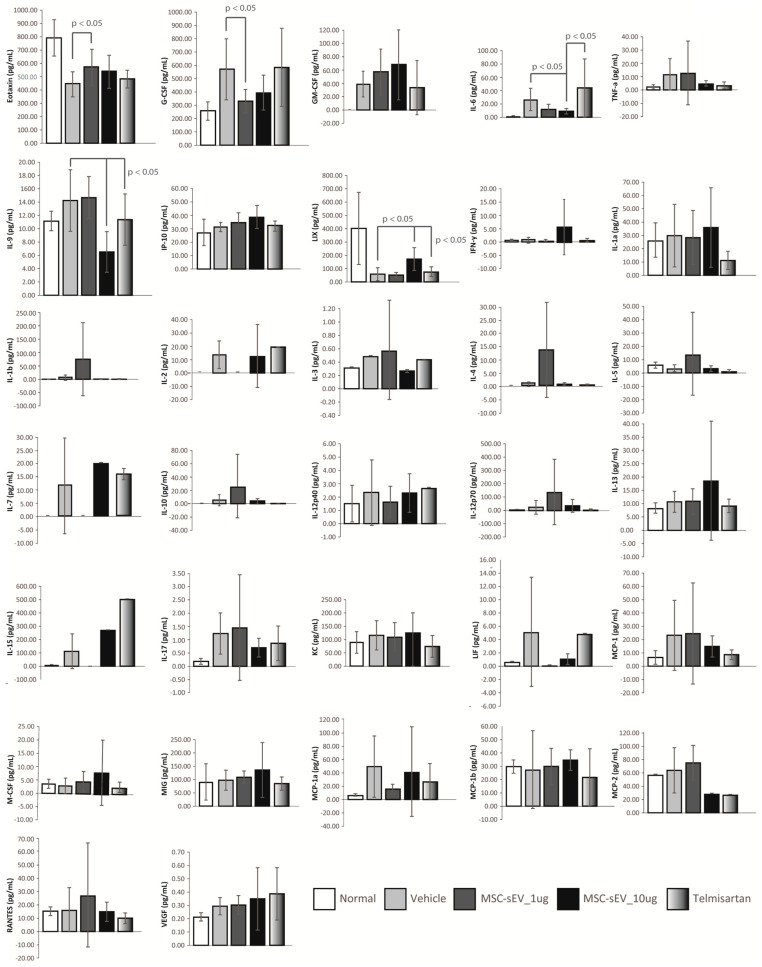
Measurement of mouse plasma cytokine/chemokine. Mice were sacrificed at nine weeks of age, and the non−fasting blood samples were collected in polypropylene tubes with anticoagulant and centrifuged at 1000× *g* for 15 min at 4 °C. The supernatant was collected and a Mouse Cytokine/Chemokine 32−Plex Discovery Assay^®^ Array was performed. The mouse plasmatic levels of Eotaxin, G−CSF, GM−CSF, IFNγ, IL−1α, IL−1β, IL−2, IL−3, IL−4, IL−5, IL−6, IL−7, IL−9, IL−10, IL−12p40, IL−12p70, IL−13, IL−15, IL−17A, IP−10, KC, LIF, LIX, MCP−1, M−CSF, MIG, MIP−1α, MIP−1β, MIP−2, RANTES, TNFα, VEGF−A were measured by Luminex technology. Statistical significance was determined by Student’s *t*−test. *p* values < 0.05 were considered statistically significant.

**Table 1 ijms-24-08092-t001:** Definition of NAFLD Activity score components.

Item	Extent	Score
Steatosis	Steatosis at 50−fold magnification	
	<5%	0
	5–33%	1
	>33–66%	2
	>66%	3
Lobular inflammation	Estimation of inflammatory foci	
	No foci	0
	<2 foci/200×	1
	2–4 foci/200×	2
	>4 foci/200×	3
Ballooning	Estimation of number of ballooning cells	
	None	0
	Few balloon cells	1
	Many cells/prominent ballooning	2

## Data Availability

The data presented in this study are all available in this article.

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
