# Peer review of "MSC−sEV Treatment Polarizes Pro−Fibrotic M2 Macrophages without Exacerbating Liver Fibrosis in NASH"

_ijms, 2023, doi:10.3390/ijms24098092_

Round 1

Reviewer 1 Report

This study focuses on the immune regulatory effects of ESC-MSC-derived exosomes in reducing liver fibrosis by inducing M2-Mac polarization. However, rather than the low novelty of this work, serious points need to be addressed before considering this work for publication. 

(1) there is no characterization for MSCs or Exos in the manuscript, such as MSCs surface markers, activity, and immune orientation. for exosomes, the size, markers, and concentration per ul. 

(2) The mechanism of exosome effects is not clear, how MSC-Exos mediated the effects on Mac polarization. The cargo of exosomes should be analyzed in terms of miRNAs, lncRNAs, proteomics, TLR markers, etc. 

(3) There is no proof that M2-Mac polarization is the key to this effect, maybe it is a result of some immunomodulation activity. 

(4) Do authors think that M2-Mac polarization is the main modulator of fibrosis reduction, what is the evidence? 

The authors need to perform further experiments to solve these issues and explain the limitations of this work. 

Reviewer 2 Report

In this study, the authors supported that MSC-sEV treatment was efficacious in reducing liver fibrosis in a mouse model of NASH despite an increase in pro-fibrotic M2 macrophage polarization. This is an interesting topic with significant prospect of clinical application. Firstly, the authors demonstrated MSC-sEV preparations can polarize M0 to M2 but not M1 macrophages in vitro. Moreover, the effect of  MSC-sEV preparations suppressing liver fibrosis in a mouse model of NASH was verified. This manuscript is a well performed study in the important role of MSC-sEV in reducing liver fibrosis of NASH. However, the manuscript requires some revisions.

1. All abbreviations in the text should be consistent. For example, in page 1 ,line 38,“MSCs” and in page 1, line 44,“MSC”; in page 1, line 20, “MSC-sEVs” and in page 2, line 54, “MSC-sEV” and in page 2, line 55 ,“MSC-EV”;

2. Please pay particular attention to English grammar, spelling and sentence structure so that the study is clear to the readers. For example, in page 5 ,line 51 ,“were” should be “was”; in page 2 ,line50, “ in the 50-200 nm size range i.e. small EVs or sEVs” should be corrected.

3. How to choose MSC-sEVs doses of 0.1, 1, 10 or 30 µg/mL to culture with the M0 macrophages in Figure 1 and Figure 2? The M0 macrophages did not increase with increasing dose of MSC-sEVs in Figure 1, please explain that.

4. How to choose the MSC-sEVs doses of 1 and 10 µg /50 µL/mouse in Figure 3? Did the mice with different weights get the same dose of MSC-sEVs ? Please state the MSC-sEVs dose of each mouse.

5. How did the authors prove that MSC-sEV reached the liver after MSC-sEV was administered intraperitoneally? I suggest that the biodistribution of MSC-sEV should be tested in mouse.

6. I suggest that the CD163 expression of Nomal group and Telmisartan group should be added in Figure 6.

7. I suggest that human ESC-derived mesenchymal stem cells and MSC-sEVs should be identified.

8. Is there a connection between polarization of pro-fibrotic M2 macrophages and the reducing of IL-6 levels in the plasma of mice? Please state that in the text.

9. The underlying mechanism of IL-6 reduction and polarization of pro-fibrotic M2 macrophages not leading to exacerbate liver fibrosis should be studied further. The following research directions are my suggestions. For example, the relationship between the two factors above and hepatic stellate cells or T and B cells the authors mentioned in Discussion.

Reviewer 3 Report

General Comments

This is a study to determine whether EVs from human-derived cells might induce inflammation and fibrosis in the livers of experimental animals.

Specific Comments

Introduction

P1, l3; “Mesenchymal stromal/stem cells (MSCs)”  This needs further definition.

P2, l49; “Therefore, it is now widely accepted that the therapeutic activity of MSCs is primarily mediated by sEVs in the 50-200 nm size range i.e. small EVs or sEVs [7]”  EVs are important for cell-cell communication, but to say that they are the primary mediators is an overstatement.

P2, l54; “MSC-sEV preparation is greatly influenced by the source and culture of the MSCs” what is meant by source?  Needs to be defined.

Materials and Methods

The authors need to provide evidence for institutional approval for human subjects providing cells for research.  They also need to provide evidence for approval for animal studies.  These issues are required for acceptance and publication.

Round 2

Reviewer 1 Report

The main aim of this study is to investigate the potential efficacy of MSCs-Exos to reduce liver fibrosis by promoting the polarization of M0 macrophage into M2 macrophage. However, there are some serious problems with the methods and results. 

(1) As seen in Fig.1A and Fig.2A the gate position is different, which is a serious problem in verifying Mac surface markers. Both figures should be from the same experiment and presented in one figure to compare the effect of your Exos, which you based on to conclude that MSCs-Exos promoted M2 but not M1 polarization. 

(2) I revised the ref provided by the authors, the characterization of MSCs was observed in Ref.1 but the validation of the Exos quality wasn't seen in both references. In addition, every project has its condition and concepts which strongly recommend proofing the quality of items used in the same project. 

(3) The important point in this project is that MSCs-Exos reduced liver fibrosis but Fig.5C showed massive fibrosis in the liver and the improvement wasn't noticed. 

(4) Finally, using MSCs-Exos for therapy without stimulating MSCs towards functional phenotype and then using the released Exos was recently discussed because released Exos should be managed and guided toward the specific function it was applied for. 

Reviewer 2 Report

There is a minor comment

To Response 7:

The identification results of other studies are not representative of ESC-derived mesenchymal stem cells and MSC-sEVs in this study. If possible, I suggest that the authors supplement the identification of ESC-derived mesenchymal stem cells and MSC-sEVs.

Reviewer 3 Report

The author correction make the manuscript acceptable.

Round 3

Reviewer 1 Report

(1) I suggest authors clarify the aim in the abstract as they explained in the response to the reviewer. 

(2) Please, remove the replication of (NAS) explanation at line no. 442.

(3) Please, add study limitations at the end of the discussion.
